# QTL Mapping for Domestication-Related Characteristics in Field Cress (*Lepidium campestre*)—A Novel Oil Crop for the Subarctic Region

**DOI:** 10.3390/genes11101223

**Published:** 2020-10-19

**Authors:** Cecilia Hammenhag, Ganapathi Varma Saripella, Rodomiro Ortiz, Mulatu Geleta

**Affiliations:** Department of Plant Breeding, Swedish University of Agricultural Sciences, 23053 Alnarp, Sweden; ganapathi.varma.saripella@slu.se (G.V.S.); rodomiro.ortiz@slu.se (R.O.); mulatu.geleta.dida@slu.se (M.G.)

**Keywords:** candidate genes, comparative genomics, domestication, *Lepidium campestre*, field cress, mapping population, oil crop, phenotype, quantitative trait loci (QTL) mapping, traits

## Abstract

Domestication of a new crop requires identification and improvement of desirable characteristics Field cress *(Lepidium campestre)* is being domesticated as a new oilseed crop, particularly for northern temperate regions.. In the present study, an F_2_ mapping population and its F_3_ progenies were used to identify quantitative trait loci (QTLs) for plant height (PH), number of stems per plant (NS), stem growth orientation (SO), flowering habit (FH), earliness (ER), seed yield per plant (SY), pod shattering resistance (SHR), and perenniality (PE). A highly significant correlation (*p* < 0.001) was observed between several pairs of characteristics, including SY and ER (negative) or ER and PE (positive). The inclusive composite interval mapping approach was used for QTL mapping using 2330 single nucleotide polymorphism (SNP) markers mapped across the eight field cress linkage groups. Nine QTLs were identified with NS, PH, SO, and PE having 3, 3, 2, and 1 QTLs, explaining 21.3%, 29.5%, 3.8%, and 7.2% of the phenotypic variation, respectively. Candidate genes behind three of the QTLs and favorable marker alleles for different classes of each characteristic were identified. Following their validation through further study, the identified QTLs and associated favorable marker alleles can be used in marker-aided breeding to speed up the domestication of field cress.

## 1. Introduction

The demand for plant-based oils is increasing as they play a crucial role in the bioeconomy by serving as a source of raw material for bioenergy, fuel production, and different types of industrial applications [1,2]. In addition, feeding a growing world population requires an increased vegetable oil production for food and feed, which makes oil crop production an important area of agricultural development [2,3].

Biennial and perennial crops provide better ecosystem services in the form of carbon storage, soil, and water managements, as compared to annual crops [4,5]. Field cress (*Lepidium campestre* (L.) R. Br), a self-pollinated biennial plant species in the family Brassicaceae [6,7], is under domestication as a biennial/perennial oilseed crop to diversify agricultural production, introduce a more eco-friendly cropping system, and meet the growing demand of plant oils [7]. Unlike any oil crop available on the market, field cress has excellent winter hardiness that allows it to grow and perform well in the colder climates of the northern temperate regions. It has proven to be very productive, with a seed yield of over 3 t ha^−1^ in areas 400 km south of the Arctic Circle (Umeå, Sweden), where winter rapeseed poorly survives (unpublished data). In more southern locations in Sweden, where winter rapeseed is the dominating oilseed crop, field cress breeding lines under sown in barley produce as high as 6 t ha^−1^ (unpublished data).

To be successful, a new crop for the subarctic region in the northern hemisphere should have a number of specific characteristics. The subarctic climate is characterized by distinct conditions, such as long, cold winters and summers with mild temperatures. At these latitudes, the crop growing season is short but has a long photoperiod during the summer, allowing the plants to grow and develop faster [8]. These agronomic conditions require crop plants that mature early and can cope with the stress of long days. For harvesting synchronously matured seeds, the plants need to stop producing new flower buds after the initial batch of flowers grow into mature seeds, thus having a determinate flowering habit. Similar to other crops, such as soybean and rapeseed [9,10], seed yield in field cress is affected by flowering behavior and pod setting. Characteristics that directly or indirectly contribute to seed yield in field cress include flowering time, number of pods per inflorescence, pod density, inflorescence length, and resistance to pod shattering. Unlike rapeseed (*Brassica napus* L.) and other brassica crops [11], increasing the number of seeds per pod as a means to increase seed yield is not an option in field cress. Instead, seed yield in field cress could be improved by increasing the number of stems, inflorescence length, and pod density through breeding. For seed crops, desirable plant height and stem growth orientation are important characteristics to avoid lodging and to facilitate harvesting. The heritability of plant height is good but, just like seed yield, it is strongly affected by environmental factors, such as photoperiod, temperature, water, and nutrient availability [12].

Field cress has a high potential to be a lucrative oilseed crop in the temperate regions with positive environmental effects [7,13,14]. The seed yield potential, possible applications of the raw materials (such as seed oil and press cake), and contribution to the improvement of agricultural practices will strongly influence the commercialization of this crop [14]. To address some of these issues and accelerate the breeding process of field cress, we are developing genomic tools for marker-aided breeding.

Field cress has the same ploidy level and chromosome number (2n = 2x = 16) with *L. heterophyllum* Benth, which can produce viable and fertile interspecific hybrids when crossed with field cress [6,14,15]. In this study, an F_2_ mapping population (and its F_3_ progenies) of an interspecific hybrid between field cress and *L. heterophyllum* (referred to as “*campestre* × *heterophyllum*” hybrid in this research article) was used to study phenotypic variation and identify quantitative trait loci (QTLs for eight agronomically important characteristics: plant height (PH), number of stems per plant (NS), stem growth orientation (SO), flowering habit (FH), earliness (ER), seed yield per plant (SY), pod shattering resistance (SHR), and perenniality (PE). This study provides novel data about QTLs associated with domestication-related characteristics in field cress and contributes to the fine mapping and identification of candidate genes responsible for number of stems per plant, plant height, and stem growth orientation.

## 2. Materials and Methods

### 2.1. Plant Material and Target Characteristics

A genotype of field cress (NO94-6-1) and a genotype of *L. heterophyllum* (Pl 597856-3b), which both show a significant variation for the target characteristics, were used as starting germplasm for this study. Seeds from NO94-6-1 and Pl 597856-3b were planted in 2.5 L plastic pots in a greenhouse at the Swedish University of Agricultural Sciences (SLU), Alnarp, Sweden, in September 2013. The plants were kept in the greenhouse under high pressure sodium (HPS) lamps with a 16/8 h (light/dark) photoperiod. The temperature was set to 21/18 °C (light/dark) with a relative humidity of 60%. The plants were moved to a vernalization chamber 60 days after planting and taken back to the greenhouse after 8 weeks of vernalization. When the plants started flowering in March 2014, single plants of NO94-6-1 and Pl 597856-3b were selected and used as a pollen recipient and donor, respectively, to generate F1 hybrids. Before cross-pollinating, the whole inflorescence of both plants were bagged at the early flowering stage. A single inflorescence of NO94-6-1 was then separated from the rest, and its flower buds (before pollen maturation) were emasculated using forceps under a binocular microscope. Immediately after emasculation, the flower buds were pollinated with a mature pollen from Pl 597856-3b plant and bagged separately with a pollination bag. The F_1_ seeds were harvested in May 2014 and then planted in the greenhouse using similar pots to generate F_2_ seeds. The F_1_ plants went through the same vernalization procedure as their parents and flowered in the greenhouse in September 2014. This was followed by selecting and bagging a single F_1_ plant just before flowering. At maturity, the F_2_ seeds of this plant were harvested (November 2014) and used to generate an F_2_ mapping population.

NO94-6-1 is derived from a population originally collected from Arlid (NO94-6) in Sweden, whereas the origin of Pl 597856-3b is a USDA-ARS gene bank accession LH Pl 597,856 that was originally collected in Spain. In terms of target characteristics, NO94-6-1 is characterized by (1) a height of about 50 cm, in average, at maturity; (2) single-stem, (3) erect stature, (4) determinate flowering, (5) late maturing, (6) about 15 gm seeds plant^−1^ on average, (7) a moderate level of pod shattering, and (8) biennial growth habit. On the other hand, Pl 597856-3b is characterized by (1) a height of ca. 30 cm tall, in average, at maturity; (2) multi-stem, (3) semi-erect stature, (4) indeterminate flowering, (5) early maturing, (6) less than 10 gm seeds plant^−1^ in average, (7) a high level of pod shattering, and (8) facultative perennial growth habit. Plants with moderate levels of pod shattering generally lose 20–30% of their seeds under environmental conditions that cause above 90% seed loss in genotypes susceptible to pod shattering. Plants that lose above 30% of their seeds under adverse environmental conditions at maturity are regarded as susceptible to pod shattering in this study. Plant pictures showing favorable and unfavorable characters of the traits studied are presented in Appendix A.

### 2.2. Planting, DNA Isolation, and Genotyping

The F_2_ mapping population, their parents, and three additional genotypes were planted in 2.5 L plastic pots, under the same greenhouse conditions as above, in May 2015. Fresh leaf tissue was sampled from young seedlings individually and DNA was extracted, as described in Gustafsson et al. [3]. The DNA samples of 380 individual plants, comprising 375 F_2_ plants, two parents, and three additional genotypes (two field cress and one *L. heterophyllum*) were sent to Cornell University for genotyping-by-sequencing (GBS) at the Genomics Facility of the Biotechnology Resource Center [16].

### 2.3. Phenotyping of Mapping Population

The 375 individual plants of the F_2_ mapping population were transferred from the greenhouse to the vernalization chamber 60 days after planting. After vernalization for 8 weeks, the plants were transferred back to the greenhouse and phenotyped for PH, NS, and SO in the greenhouse (Table 1). Sufficient spacing between the F_2_ plants was applied to avoid potential cross-pollination through direct contact between plants. At plant maturity, PH was measured in cm, NS was counted, and SO was visually determined. Each F_2_ plant was then harvested individually in October 2015. In August 2016, the harvested F_3_ seeds were planted in 2.5 L plastic pots in the same greenhouse for further study. Each F_2_ plant was represented by six F_3_ plants. At 2 months after planting (October 2016), the 2250 plants were transplanted to two field sites in Alnarp for overwintering. A total of 1 m spacing between plants was applied. At each site, each F_2_ plant was represented by three F_3_ plants. After overwintering, the plants flowered during the spring of 2017. Phenotypic data were collected for FH, ER, SY, SHR, and PE (Table 1) during the summer and autumn of 2017. Second round PE data was collected during April–May 2018. FH was recorded as determinate, moderately indeterminate, and indeterminate. In terms of the time it takes to reach full maturity after overwintering (earliness), the plants fall into two clear categories. The early maturing group matures within 90–100 days, whereas the late maturing group needs 115–125 days to mature. The amount of seeds produced by each plant (SY) were categorized into four classes, with the low yielding group producing less than 5 gm of seeds plant^−1^, whereas the high yielding group produced more than 20 gm seeds plant^−1^ (Table 1). Data for SHR was collected three times after the plants reached full maturity and were ready for harvesting. The three data sets on SHR were collected shortly after heavy rain and strong winds within 2–4 weeks after full maturity of the plants. The final SHR data was collected immediately after the susceptible checks lost more than 90% of their seeds. The percentage of shattered pods was calculated after the number of intact pods and shattered pods were recorded for each plant. In the case of perenniality, plants were grouped into biennial, facultative perennial, and perennial after collecting data on (1) regrowth (ratooning) of the plants during autumn 2017 and (2) the winter-survivability of the ratoons during the spring of 2018. Biennial plants fully died out after the first harvest. Facultative perennials developed ratoons that grew well during the autumn but did not survive the winter. Perennials regrew well after the first harvest, overwintered, and harvested for the second time the year after.

### 2.4. Phenotypic Data Analysis

Phenotypic data of the eight characteristics were analyzed using Minitab^®^ 18.1 (2017). These included analysis of the Pearson correlation coefficient among data collected from the two sites in the F_3_ population, characteristics recorded in the F_2_ population, and characteristics recorded in the F_3_ population, as well as characteristics recorded in the F_2_ and F_3_ populations.

### 2.5. SNP Data and QTL Analysis

The analysis of GBS data generated at the Genomics Facility at the Biotechnology Resource Center of Cornell University produced 120,438 biallelic SNPs, of which 2330 SNPs were mapped to eight linkage groups, as described in Geleta et al. [16]. These mapped SNPs were selected and used for QTL analysis in this study (see Appendix A at https://www.frontiersin.org/articles/10.3389/fpls.2020.00448/full#supplementary-material). The QTL analysis was performed using QTL IciMapping software v4.1 [17] for the eight characteristics collected from F_2_ and corresponding F_3_ populations. The previously developed genetic map with eight linkage groups [16] were further processed using in-house scripts to generate an input file (in *.bip format) for QTL analysis, together with the phenotypic data set. QTL mapping was performed using Inclusive Composite Interval Mapping (ICIM) [18] with the Inclusive Composite Interval Mapping of Additive and Dominant (ICIM-ADD) QTL method [17], whose default settings include the Kosambi mapping function, a *p* value of 0.001 as the probability for entering variables (PIN), and a 1 cM scanning step. The score threshold for the logarithm of the odds (LOD) was 3.

### 2.6. Comparative Genomics/Candidate Gene Identification

Following the identification of QTLs, the positions of the SNP markers flanking the QTLs were used as references to further analyze the flanked genomic regions harboring the newly identified QTLs. The DNA sequence within each pair of flanking SNPs was BLAST searched against the *Arabidopsis thaliana* genome in the National Center for Biotechnology Information (NCBI) databases to identify homologous regions. The homologous regions of *A. thaliana* were then examined for potential candidate genes related to the characteristics targeted in this study. 

### 2.7. SNP Genotypes Versus Trait Variation

The association between the three genotypic classes of each SNP flanking a QTL and the phenotypic classes of the corresponding characteristic was analyzed, and the relationship between the genotypes (designated AA, AB, and BB) at each flanking SNP locus and the average phenotypic value of the corresponding characteristic were determined. Analysis of variance (ANOVA) was conducted using Minitab 18 to determine the statistical significance of the association.

## 3. Results

### 3.1. Analysis of Phenotypic Data of F_2_ and F_3_ Populations

Eight characteristics (PH, NS, SO, FH, ER, SY, SHR, and PE), considered to be very important in the domestication of field cress were evaluated in the F_2_ or F_3_ populations. SO, FH, ER, SY, SHR, and PE were treated as categorical variables, whereas PH and NS were recorded as quantitative variables. The description for all characteristics and the proportion of each category in each categorical characteristic in the F_2_ or F_3_ population are presented in Table 1, and photos of plants with desirable and undesirable characteristics of the traits are displayed in Appendix A. The PH ranged from 10 to 49 cm (mean 26.6 cm) and the NS ranged from 1 to 15 (mean 4.9) in the F_2_ population. In the case of SO, the majority of the F_2_ plants (53.1%) had an erect central stem, while that of the remaining plants were semi-erect or creeping type (38.5% and 8%, respectively). FH refers to the ability of a plant to continue flowering after the first group of inflorescences reaches maturity. For this characteristic, the plants of the F_3_ population were of the determinate (22%), the moderately indeterminate (34%), or, the slightly more common, indeterminate (43%) type. For ER, plants of the F_3_ population were categorized as the early or late type and the vast majority were the early type (86%). The seed yield in individual plants of the F_3_ population was categorized into four classes (low, medium, medium-high, and high), and the medium and low were the dominant classes, accounting for 35.7% and 52.9% of the plants in the population, respectively (Table 1). Referring to SHR, the majority of the plants in the F_3_ population were susceptible to pod shattering (87%) and only a few plants were highly resistant against pod shattering (0.7%). In the case of PE, most of the plants in the F_3_ population were regarded as biennial (70.5%), while a minority were either facultative perennial (13.9%) or perennial (15.6%). Erect stems and early maturity are considered desirable in the domestication of field cress, and plants with these characters were dominant in the populations investigated (Table 1).

The Pearson correlation coefficient between site-1 and site-2 was calculated for each category of each characteristic, as well as for the combination of all categories of each characteristic. Significant positive correlations were found between the two sites for all categories, except for “moderately indeterminate” of FH and “high resistance” of SHR (Table 2). The correlation analysis at a characteristic level showed a highly significant positive correlation between the two sites for all characteristics. The highest positive correlation recorded between the two sites was for combined data of SHR (0.80), followed by ER (0.70) and PE (0.56) (Table 2).

Among the three characteristics recorded in the F_2_ population, highly significant positive correlations were observed between NS and PH, whereas SO showed significant negative correlation with PH, reflecting that erect plants tend to be taller than semi-erect and creeping types (Table 3). Highly significant positive correlations were observed between the following pairs of characteristics measured in the F_3_ population: SHR and SY, ER and FH, and PE and ER. On the other hand, SY showed a highly significant negative correlation with ER, FH, and PE. Similarly, ER and SHR showed a highly significant negative correlation. The highest negative correlation was observed between ER and SY (r = −0.66). NS and SO showed a significant correlation only with PH (Table 3). There was no significant correlation between characteristics measured at F_2_ and F_3_, except in the case of PH vs. FH. PH, which showed a significant positive correlation with determinate flowering, thus indicating that taller plants showed a higher tendency to have determinate flowering than shorter plants.

Among the 2250 plants of the F_3_ population planted at the two field sites, 1656 plants (73.6%) had complete data for all target characteristics. Phenotypic data from these 1656 individual plants were analyzed to identify plants possessing one or more desirable characters of the five target characteristics. These were: determinate flowering (for FH), early maturity (for ER), high pod shattering resistance (for SHR), large plants with high seed yield (for SY), and perennial type (for PE) (Table 1, Figure 1). The majority of the plants analyzed (57%) had only one of the desirable characteristics, where early maturing types (EMT) were the most common (53.3%) and highly shatter-resistant (HSR), perennial type (PT), large plants (LPs), and plants with determinate flowering (DF), accounted for 3.7% together (Figure 1a). About 10.3% of the plants lacked desirable characteristics for all five characteristics, whereas about one third of the population (534 plants; 32.8%) showed more than one desirable characteristic. Among plants having more than one desirable characteristic, those that combined determinate flowering and early maturity were the most common (51.9%), followed by those combining early maturity and perenniality (29.8%). Plants that combined determinate flowering, early maturity, and perenniality accounted for 15.3% of the 534 plants (Figure 1b).

### 3.2. QTL Analysis

QTL analysis revealed a total of nine QTLs (LOD score >3) for four of the eight characteristics analyzed. The QTLs were distributed across seven of the eight linkage groups (LGs) (Table 4; Figure 2). LG4 and LG6 had two QTLs each, while LG1, LG2, LG3, LG7, and LG8 had one QTL each. Three of the QTLs (on LG1, LG6, and LG8) are for NS, which collectively accounted for 21.3% of the observed variation in this characteristic. The most important of these QTLs is *qNS1-1* that accounted for 11.8% of the observed variation (Table 4). Similarly, three QTLs were identified for PH (on LG3, LG6, and LG7). These QTLs collectively accounted for 29.5% of the characteristic variation, with *qPH6-1* accounting for 16.9%. For SO, two QTLs were found (on LG2 and LG4), which, together, had a phenotypic effect of 3.8%. A QTL for PE (on LG4) accounted for 7.2% of the observed variation in perenniality (Table 4).

### 3.3. Comparative Analysis of the Genomic Regions Harboring the QTLs Using A. thaliana Genome

About 88% of the sequences of field cress linkage groups show synteny with corresponding *Arabidopsis* chromosomes [16], which makes comparative genomics with *A. thaliana* a powerful tool for analyzing the QTL regions of field cress. Genomic regions in *A. thaliana* chromosomes homologous to field cress genomic regions harboring the QTLs were identified through comparative genomic analysis for all QTLs, except for *qSO2-1* and *qPE4-1*. These last two QTLs do not have homologous regions in *A. thaliana* genome (Table 4). For the seven QTLs, the homologous regions in *A. thaliana* chromosomes were screened for candidate genes known to regulate the characteristics targeted in this study. Several candidate genes were detected for QTLs related to SO, NS, and PH (Figure 3). The homologue of the *POLTERGEIST* (*POL*) gene, which regulates meristem development through interactions with *CLAVATA1* (*CLV1*) and *CLAVATA3* (*CLV3*) in *A. thaliana* [19], is a likely candidate gene behind *qSO4-1*. *qNS6-1* corresponds to the homologous region that contains three *XYLOGLUCAN ENDOTRANSGLUCOSYLASE/HYDROLASE* genes, *XTH12, XTH13*, and *XTH25*, which are known to regulate brassinosteroid biogenesis in *A. thaliana* [20], suggesting that their homologues are a likely gene behind *qNS6-1*. Homologue of *BROX1*, which is also a known regulator of brassinosteroids (hormones that are known to promote stem elongation and affect other aspects of plant growth) in *A. thaliana* is a likely gene behind *qPH7-1.*

### 3.4. SNP Genotypes Versus Trait Variation

The relationship between the genotypes of the linked markers closest to the QTLs on both sides and the average phenotypic values was analyzed to assess the association between the actual segregation of the SNP markers in relation to the phenotypes of a target characteristic (Figure 4). Among the three QTLs, for number of stems per plant (NS), *qNS1-1* is the strongest, accounting for 11.8% of the phenotypic variation (Table 4). The marker loci flanking this QTL are sc26535_625A/G and sc21796_243C/T (Figure 4a), and alleles designated as “B” at these loci are favorable for multiple stems per plant, whereas “A” alleles are favorable for producing single-stem plants. The average NS for BB homozygotes was 6.5, whereas it was 3.8 for AA homozygotes at both loci. The difference between the two genotypes was statistically highly significant (Figure 4a). Among the plants with ≥10 stems/plant, 46% and 13% were BB and AA homozygotes, respectively, at sc26535_625A/G locus (Table 5). At the same locus, AA and BB genotypes accounted for 56% and 8% of all single-stem plants. Considering all three QTLs for NS together, the AA + AA + AA homozygotes at the three left marker loci had an average of 2.5 stems plant^−1^, whereas BB + BB + BB homozygotes had eight stem plants^−1^ on average.

The following are the pair of marker loci flanking the three QTLs for PH: *qPH3-1* (sc11378_1260A/G and sc9980_604A/T_989T/G), *qPH6-1* (sc28125_3971C/A and sc26361_413G/C), and *qPH7-1* (sc21252_1620C/T and sc20822_9341C/T). Among the three QTLs, *qPH6-1* was the strongest, with 16.9% phenotypic effect (Table 4). The average PH for the different genotypes of these marker loci varied from 20.3 cm to 31.1 cm. Favorable alleles for tallness at marker loci linked to *qPH3-1*, *qPH6-1*, and *qPH7-1*, were designated as “A”, “B”, and “B”, respectively (Figure 4b). For example, among plants that were above 30 cm tall, 48% were BB homozygotes, whereas only 3% were AA homozygotes at sc28125_3971C/A locus, which is the left flanking marker for *qPH6-1* (Table 5). At the same locus, AA and BB genotypes accounted for 38% and 11% of plants that were shorter than 20 cm. Considering the three QTLs for PH together, the average height of plants that were homozygous for favorable alleles at the three left marker loci (AA + BB + BB) was 34 cm, whereas those that had BB + AA + AA genotype were 11 cm. Among plants with AA + BB + BB genotype, 78% were taller than 30 cm.

In the case of stem growth orientation (SO), the pair of marker loci flanking *qSO2-1* are sc33009_557C/A and c1443202_1111C/G, whereas that of *qSO4-1* are sc9980_604A/T and sc3610_484A/G (Figure 4c). For having erect stems, favorable alleles at the marker loci linked to *qSO2-1* and *qSO4-1* are designated as “B” and “A”, respectively. At sc33009_557C/A locus, the AA and BB genotypes accounted for 15% and 38% of all erect plants, whereas 37% and 18% of erect plants had AA and BB genotypes at sc9980_604A/T locus (Table 5). Considering the two QTLs together, 77% of the plants with BB + AA genotype at the two left marker loci were erect. The flanking marker loci for *qPE4-1*, a minor QTL for perenniality, are sc2646_16112C/A and sc16365_3162C/T. Favorable alleles at these loci for perennial life form is designated as “B”. Among the F_3_ plants phenotyped, only 15.7% were perennial. Among the perennial plants, those that are homozygous for “A” and “B” alleles at sc2646_16112C/A locus accounted for 14% and 34%, respectively (Figure 4d).

Analysis of variance (ANOVA) conducted by combining all QTLs for each characteristic revealed that the percentage of plants that are homozygous for favorable marker alleles are significantly higher than the percentage of plants that are homozygous for unfavorable marker alleles for (1) NS; i.e., having 10 or more stems per plant (*p* < 0.01) or having a single stem (*p* < 0.05), (2) PH, being taller than 30 cm (*p* < 0.05) or being shorter than 20 cm (*p* < 0.01), (3) SO, being erect (*p* < 0.01) or creeping (*p* < 0.01), and (4) PE, being either perennial (*p* < 0.01) or biennial (*p* < 0.05) (Table 5).

## 4. Discussion

Domesticating a novel crop involves characterization, evaluation, and improvement of various agro-morphological characteristics, such as those investigated in this study. The identification of genomic regions that control these characteristics is a significant contribution for implementation of marker-assisted breeding in field cress. This study focused on eight characteristics that are of high importance for the domestication of field cress and identified nine QTLs distributed over seven linkage groups, contributing to the number of stems per plant, plant height, stem growth orientation, and perenniality. Five interesting candidate genes, encoding proteins implicated in plant structure regulation, were co-located, with the QTL affecting PH, SO, and NS. Furthermore, SNPs that could be used for marker-aided breeding after validation in the future crossbreeding experiments were identified.

The short growing season of northern Sweden necessitates fast maturing crops; hence, developing early maturing types is of the highest importance as field cress is targeted for this area. A majority of the plants in the F_3_ population were categorized as early maturing types, and plants with this desirable characteristic tended to be determinate flowering type, as shown with a highly significant positive correlation between early maturity and determinate flowering (Table 3). On the other hand, early maturing types produced low seed yield on average, as high seed yield was negatively correlated with early maturity. In line with this, late maturing types of field cress have had a relatively higher seed yield than early maturing types, on average, as shown in a number of field trials (unpublished data), which may suggest a trade-off between early maturity and high seed yield. Such negative correlations between these characteristics have been reported in different crops, including wheat [21] and cowpea [22].

The height of mature wild field cress plants vary between 20 and 60 cm [23]. In this study, plant height varied between 10 and 49 cm, with a modal value of 30 cm in the F_2_ population. This characteristic showed a highly significant positive correlation with NS. Our correlation analysis showed that erect plants were taller than other forms of stem growth orientation, which is as good as both taller height and erectness, which are considered desirable in field cress domestication. Three QTLs were identified for PH and were collectively responsible for 29.5% of the observed variation. Analyzing the effects of *qPH6-1* and *qPH7-1* on phenotypic differences showed that the BB genotypes for the closest linked SNP markers are associated with a plant height above 30 cm, while the AA genotype is associated with a plant height below 20 cm. Plant height is a complex dynamic characteristic that is regulated through the interactions of many genes that may behave differently during different growth stages. Thus, different QTLs associated with plant height may be detected by phenotyping during specific growth periods. For plant height, the B alleles linked to *qPH6-1* is the most important for selecting taller plants among the *campestre* × *heterophyllum* hybrids and may be in other *Lepidium* populations with different genetic backgrounds. Our analysis showed that 78% of plants with the AA + BB + BB genotype for the three left flanking markers were taller than 30 cm. The results obtained in this study clearly showed that selecting based on favorable alleles at the marker loci flanking the three QTLs for PH is an efficient approach for *campestre* × *heterophyllum* derived lines. Application of these markers on wider field cress breeding materials will help determine the significance of using these markers for marker-aided breeding.

The number of stems per plant (NS) in the F_2_ population varied from 1 to 14 with a mean of 5.2. However, of all the analyzed characteristics, NS was only significantly correlated to PH. Three QTLs on LG1, LG6, and LG7 were identified for NS and collectively explained about 20% of the observed variation. The analysis of the effect of *qNS1-1* demonstrated that the BB genotype is associated with a higher number of stems (average 6.5) compared to the AA genotype (average 3.8). Considering the three QTLs for NS, the BB + BB + BB genotype had the highest NS (8 stems/plant) when compared to that of all other homozygous genotypes at these loci, such as AA + BB + BB. The result suggests strong additive effects of these QTLs. A repeated field trial is required to determine if having multiple stems/plants is an advantage in terms of seed yield in field cress. However, it is obvious that plants with a larger number of stem plants^−1^ are advantageous to fill in gaps between plants in the case of sparse sowing and when plant density is low for various reasons. During the last 7 years, we have made several interspecific crosses between different genotypes of *L. campestre* and *L. heterophyllum*. All of them have similar patterns of variation at the F_2_ generation, with regard to the characteristics targeted in this study. Several of our advanced breeding lines within the *Lepidium* domestication project are derived from *campestre* × *heterophyllum* hybrids. These lines possess various interesting characteristics that are lacked by lines derived, based only on crossbreeding of different genotypes of *L. campestre*, such as strong perennial growth habit and very high pod shatter resistance [14]. Hence, the markers at the flanking loci of the QTLs identified in this study can assist the selection of genotypes with desirable characteristics, particularly at the early generations of crossbreeding. The first QTL for NS, *qNS1-1*, may be regarded as a major QTL as its effect is higher than 10%. Selecting BB homozygotes at sc26535_625A/G and/or sc21796_243C/T loci (flanking *qNS1-1*) among *campestre* × *heterophyllum* hybrids increases the efficiency of selecting plants with multiple stems/plants (≥10) by more than six-fold and threefold when compared to selecting samples randomly and based on AA homozygotes, respectively. On the other hand, if a single stem plant^−1^ is preferred, selecting AA genotypes increases the selection efficiency by several fold, as compared to selecting samples randomly or based on BB homozygotes. Hence, the markers at these loci can be used for marker-aided selection.

More than half of the F_3_ population had an erect stem. Two minor QTLs for SO were identified on LG2 and LG4, respectively, each of which accounted for about 2% of the phenotypic variation in the population. The positive effect of *qSO2-1* on the difference in stem growth orientation is demonstrated, since a BB genotype is more than two times more likely to have an erect stem than an AA genotype at the left flanking marker locus sc33009_557C/A. Similarly, plants with an AA genotype at marker loci sc9980_604A/T (linked to *qSO4-1*) are twice more likely to have erect stems than BB genotype qSO4-1. Thus, these SNPs could potentially be used as indicators for stem growth orientation in field cress domestication and breeding. In the F_2_ mapping population used in this study, 54% of the plants had erect stems. On the other hand, 77% of all plants that were homozygous for favorable alleles at the marker loci linked to the two QTLs had erect stems. Hence, marker-assisted selection, based on the favorable alleles at these marker loci, can increase the efficiency of selection, although the two QTLs were minor, with phenotypic effects of about 4%.

There was a low but significant correlation between the two field sites (r = 0.13) for perennial type plants. The low correlation suggests strong genotype by environment interactions in this characteristic. A well-known strategy of many plants to cope with the unpredictability and variability of the surrounding environment is to switch between being an annual and perennial [24]. The results from this study and our other unpublished work showed flexibility in the life cycle, at least in certain genotypes of field cress, depending on the prevailing conditions. The results from the correlation analysis between the characteristics showed that there was a highly significant negative correlation between perennial growth habit and high seed yield. Perennial crops usually yield lower than annual crops, which is often explained as a trade-off for having a longer life cycle [25,26]. Perennial plants can distribute their seeds and offspring over multiple years, while an annual plant needs to produce all its seeds in one season, resulting in higher yields from annual crops. Another characteristic that showed highly significant correlation with perennial growth habit was earliness, although the correlation was positive. The vast majority (93%) of plants in the F_3_ population with a perennial growth habit were early maturing types. One QTL was identified for PE, explaining 7.2% of the observed phenotypic variation. Research has shown that perennial seed crops are preferable, in terms of ecosystem services, than annuals and biennials, as they allow better energy use efficiency, carbon storage, and water and soil management [4,27,28]. One of the major objectives of field cress domestication is the development of perennial cultivars in parallel with the biennials [14]. Perenniality (PE) is one of the target characteristics in this study. *L. heterophyllum* is a facultative short-lived perennial, closely related to and easily crossable with the biennial field cress [6,14,15]. Several *campestre* × *heterophyllum* hybrids have been generated within the field cress domestication project, and the proportion of perennating plants within F_2_ populations derived from different parental pairs were as high as 30%. However, few of them led to strongly perennial transgressive segregants that produced satisfactory seed yield up to four times [14]. Within the F_3_ population used in the present study, only 15.7% were perennial. However, 34% of individuals with BB genotypes at the left flanking SNP locus (sc2646_16112C/A) in this population were perennial. Hence, the efficiency of selecting potentially perennial plants can be increased through selecting favorable alleles at sc2646_16112C/A and sc16365_3162C/T marker loci linked to *qPE4-1*.

Interestingly, four of the five identified candidate genes behind two of the QTLs are regulators of the brassinosteroid biosynthetic pathways. Impairment of the brassinosteroids synthesis or signaling pathway leads to a dwarf phenotype, while increased levels of brassinosteroids lead to increased plant size [29]. In fact, brassinosteriods have been pointed out as a critical signal for the regulation of plant growth, thus having a great agronomic potential for marker-aided selection and as gene editing targets [30]. In the syntenic region of *qPH7-1, A. thaliana* contains the *POL* gene. *POL* has been shown to be a partial suppressor of meristem defects in clv1 and clv3 mutants [19]. It has also been reported that *POL* and *POLTERGEIST LIKE 1* (*PLL1*) are key players in regulating stem-cell differentiation and are the closest known factors to *WUSHEL* (*WUS*) regulation in the shoot meristem [31]. *WUS* is the main checkpoint for stem cell control and also regulates *CLV3* expression [32,33]. During ongoing re-domestication of the *Solanaceae* orphan crop, groundcherry (*Physalis pruinosa)*, it was reported that mutations in the *CLV-WUSHEL* pathway lead to increased floral meristem size, additional flower organs, and larger fruits [34]. This suggests that the homologue of *POL* is a potential candidate gene behind *qPH7-1*, which contributes to the regulation of plant height in field cress. Further investigation of the homologues of *POL* and other potential candidate genes is needed to confirm their role behind the QTLs identified in this study.

Among the 1656 F_2:3_ plants analyzed, 32.8% had more than one desirable characteristics, among which those having two and three desirable characteristics accounted for 84% and 16%, respectively. The obtained results suggest that determinate flowering and early maturity are probably the easiest desirable characteristics to combine in field cress breeding, followed by early maturity and perennial growth habit. Combining these three characteristics can also be achieved fairly easily. However, plants that combined other desirable characteristics were either few or absent in this mapping population. Hence, developing *campestre* × *heterophyllum* hybrid derived lines that combine multiple desirable characteristics, such as determinate flowering, early maturity, pod shatter resistance, and high seed yield, may be challenging and requires several generations of breeding efforts.

## 5. Conclusions

Fast maturing crop cultivars are required for cultivation in the high-latitude northern temperate regions that have short growing seasons. Hence, developing early maturing field cress is of high importance as the plant is targeted to grow mainly in these regions. The significant positive correlations observed between early maturity, determinate flowering, and perenniality suggests that genotypes combining these desirable characteristics can easily be achieved through breeding. However, given that high seed yield was negatively correlated with early maturity and perennial growth habit, increasing the seed yield of early maturing perennial field cress could be challenging and may require screening of diverse germplasm and multiple rounds of crossbreeding and selection. None of the plants analyzed had desirable characteristics of all characteristics included in this study, entailing the complex process of crop domestication. Hence, developing several breeding lines with different desirable characteristics separately, followed by their combination into common breeding lines through crossbreeding, could be a preferred strategy in field cress domestication. The nine QTLs identified in the present study will have significant contribution towards the domestication process. Particularly, the QTLs for NS and PH were the most important findings of this study, as they collectively explained over 20% of the observed phenotypic variation in the two characteristics. The candidate genes identified through comparative genomics analysis was another interesting result, and the analysis of their variation in relation to the phenotypic variation of corresponding characteristics will be an important step towards their confirmation as genes behind the QTLs. The use of favorable marker alleles identified in this study for selection of plants with desirable characters can increase the efficiency of selection and the overall pace of the domestication process. However, the findings of this study need to be validated through further research using germplasm with different genetic backgrounds, grown in diverse environments for their use in the overall marker-aided domestication process.

## Figures and Tables

**Figure 1 genes-11-01223-f001:**
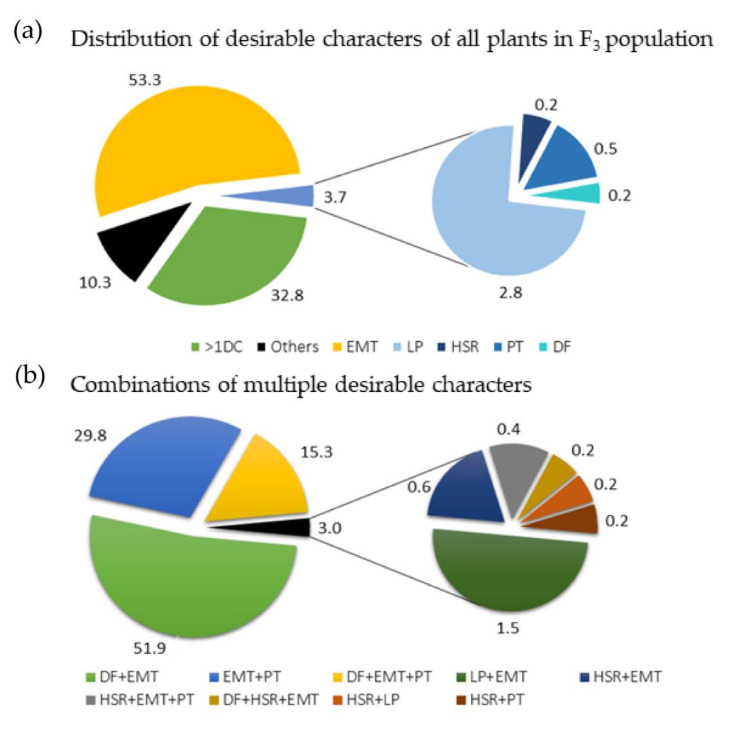
A total of 1656 F_3_ plants were analyzed for the five characteristics: flowering habit, pod shattering resistance, seed yield per plant, earliness, and perenniality. (**a**) Pie charts showing the percentage of plants having no, one, or more than one desirable characters of the five characteristics among the 1656 F_3_ plants. (**b**) Pie charts showing the percentage of plants, showing different combinations of desirable characters for the five characteristics among 543 plants (32.8% of the 1656 plants analyzed), showing more than one desirable characteristic. >1DC = have more than one desirable characteristic; early maturing type (EMT); large plants (LPs); highly shatter-resistant (HSR); perennial type (PT); determinate flowering (DF); Others = lack desirable characters. Note: for plants having more than one desirable characteristic, “+” is used to show the combination; e.g., DF + EMT = the plants combine determinate flowering and early maturity (early maturing plants with determinate flowering).

**Figure 2 genes-11-01223-f002:**
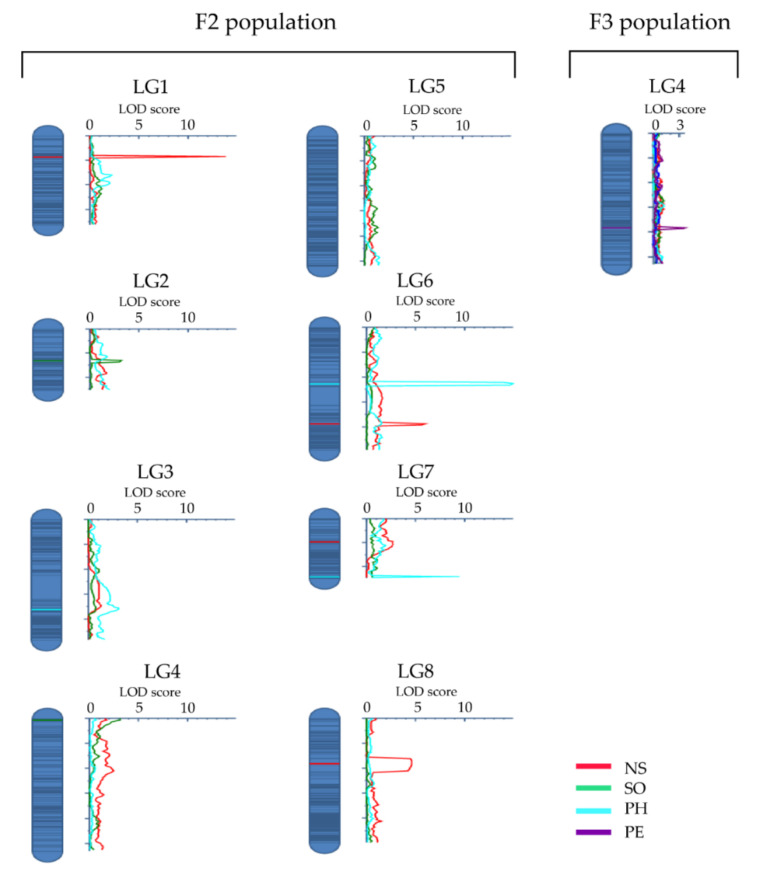
The distribution of the nine QTLs identified in this study across the field cress linkage groups for four characteristics: number of stems per plant (NS, red), stem growth orientation (SO, green), plant height (PH, turquoise), and perenniality (PE, purple). The QTLs are identified based on the F_2_ population data, except the QTL for PE on LG4, which was identified based on the F_3_ population data. The *x*-axis shows logarithm of the odds (LOD) score values.

**Figure 3 genes-11-01223-f003:**
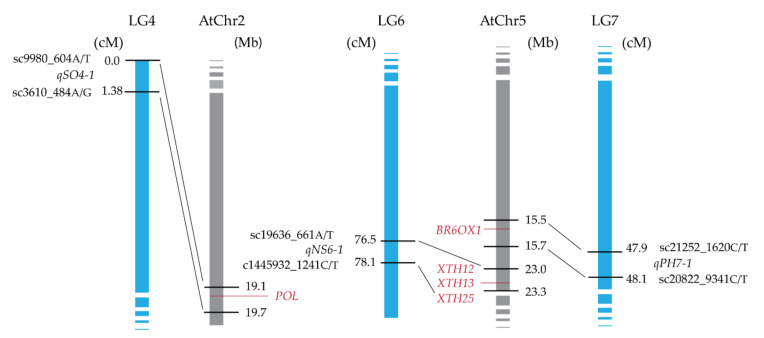
Candidate genes for the QTLs, controlling stem growth orientation (*qSO4-1*), number of stems per plant (*qNS6-1*), and plant height (*qPH7-1*) in field cress. A comparison of the homologous regions between flanking markers from *L. campestre* linkage groups (LGs) and *A. thaliana* chromosomes (AtChr2 and AtChr5). The position of flanking markers is given in cM on *L. campestre* LGs and in Mbp on *A. thaliana* chromosomes. *A. thaliana* homologous genes of interest are shown in red.

**Figure 4 genes-11-01223-f004:**
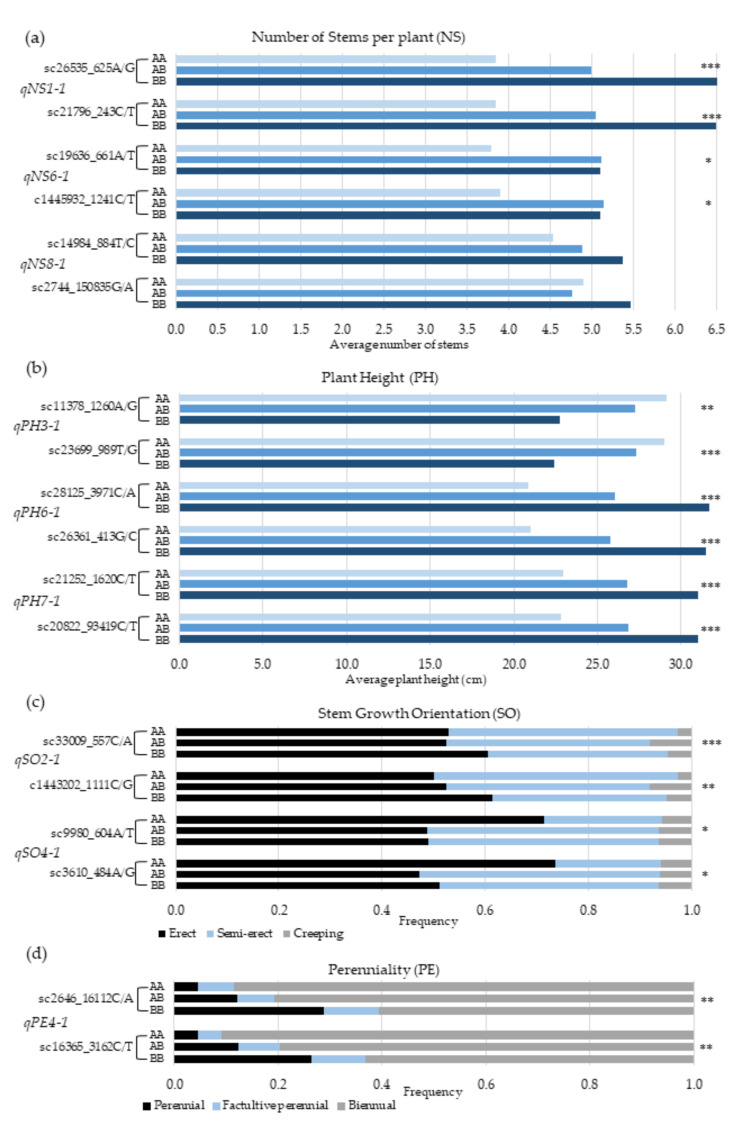
QTL effects expressed as differences in phenotypes for each genotype of each locus. (**a**) QTL effects expressed as differences in average number of stems per plant (NS) for each genotype of the linked markers, flanking the three QTLs on each linkage group (LG1, LG6, and LG8. (**b**) QTL effects expressed as differences in average plant height (PH) in cm for each genotype of the markers, flanking the QTLs on LG3, LG5, and LG7. (**c**) QTL effects expressed as differences in frequency of erect, semi-erect, and creeping stem growth orientation (SO) for each genotype of the markers, flanking the two QTLs on LG2 and LG4. (**d**) QTL effects expressed as differences in frequency of perennial, facultative perennial, or biennial types of life cycle (PE) for each genotype of the markers, flanking the QTL on LG4. The asterisks denote the statistical significance level of the variance (* = 0.05 < *p* <0.01; ** 0.01 < *p*<0.001; *** = *p* < 0.001).

**Table 1 genes-11-01223-t001:** Description of characteristics analyzed in this study: (1) plant height (PH), number of stems per plant (NS), and stem growth orientation (SO) were determined for the 375 individual plants of the F_2_ population grown in the greenhouse; (2) flowering habit (FH), earliness (ER), seed yield per plant (SY), pod shattering resistance (SHR), and perenniality (PE) were determined for the 2250 individual plants of the F_3_ population grown at two field sites (1125 plants at each site). The percentages of each category measured at F_2_ or F_3_ stages is given in the fourth column for the categorical characteristics. Desirable characters of the characteristics for field cress domestication are shown in bold.

Characteristic	Description	Measurement/Count/Category	Percentage of Each Category as Measured in F_2_ or F_3_ Population
PH ^a^	Height of each plant measured in cm at maturity	measurement in cm	NA
NS ^b^	Number of basal branches (tillers) of each plant at maturity	count	NA
SO	Growth orientation of the central stem with reference to the surface	**1 = erect** (**ca. 90°**) 2 = semi-erect (≥45°) 3 = creeping (<10°)	53.1 38.5 8
FH	The ability of plants to continue flowering after the original group of inflorescence matures	**1 = determinate** 2 = moderately indeterminate 3 = indeterminate	22 34 43
ER	Determined based on number of days from end of winter to plant maturity	**1 = early** (**90–100 days**) 2 = late (115–125 days)	86 14
SY	Determined based on the weight of seeds harvested from each plant	1 = low (<5 gm) 2 = medium (5–10 gm) 3 = medium-high (11–20 gm) **4 = high** (**>20 gm**)	52.9 35.7 8.1 3.3
SHR ^c^	Determined based on the percentage of seeds lost at full maturity after exposure to heavy rain and strong wind	**1 = high resistance** (**<10% seed loss**) 2 = moderate resistance (10–30% seed loss) 3 = susceptible (>30% of seed loss)	0.7 12.3 87
PE	The ability of the plants to ratoon and growth to full maturity after first harvest	**1 = perennial **** 2 = facultative perennial *** 3 = biennial	15.6 13.9 70.5

^a^ PH varied from 10 to 49 cm with a median, mode, and mean of 28, 30, and 26.6 cm, respectively. ^b^ NS varied from 1 to15 with a median, mode, and mean of 5, 5, and 4.9, respectively. ^c^ SHR was measured 2–4 weeks after each plant reached full maturity and was exposed to rain and strong wind that caused more than 90% pod shattering in susceptible genotypes; ** = the second harvest was made a year after the first harvest; *** = the ratoons did not survive the winter (such ratoons can grow well and be harvested under greenhouse condition). NA = not applicable.

**Table 2 genes-11-01223-t002:** Pearson correlation coefficients between site-1 and site-2 for the five characteristics ( flowering habit (FH), earliness (ER), seed yield per plant (SY), pod shattering resistance (SHR), and perenniality (PE)) recorded in the F_3_ population, based on frequencies of each category and each characteristic as a whole (combined).

Characteristic	Category	r	*p*-Value
FH	determinate	0.24	<0.001
	moderately indeterminate	0.06	0.340
	indeterminate	0.27	<0.001
	combined	0.23	<0.001
ER	early	0.31	<0.001
	late	0.31	<0.001
	combined	0.70	<0.001
SY	high	0.31	<0.001
	medium-high	0.19	0.001
	medium	0.16	0.006
	low	0.26	<0.001
	combined	0.47	<0.001
SHR	high resistance	0.01	0.808
	moderate resistance	0.25	<0.001
	susceptible	0.25	<0.001
	combined	0.80	<0.001
PE	perennial	0.13	0.037
	facultative perennial	0.13	0.026
	biennial	0.26	<0.001
	combined	0.56	<0.001

**Table 3 genes-11-01223-t003:** Pearsons correlation coefficient between the eight characteristics studied in the F_2_ population or in its F_3_ progenies.

	PH	NS	SO	FH	SY	SHR	ER
NS	0.26 ^a^***						
SO	−0.15 ^a^**	0.02 ^a^					
FH	0.12 ^b^*	0.02 ^b^	0.02 ^b^				
SY	0.11 ^b^	0.004 ^b^	0.02 ^b^	−0.34 ^c^***			
SHR	0.09 ^b^	0.09 ^b^	0.05 ^b^	−0.04 ^c^	0.18 ^c^**		
ER	0.05 ^b^	0.06 ^b^	0.02 ^b^	0.37 ^c^***	−0.66 ^c^***	−0.26 ^c^***	
PE	0.02	0.09 ^b^	0.03 ^b^	0.14 ^c^*	−0.42 ^c^***	−0.07 ^c^	0.25 ^c^***

* = 0.01 < *p* < 0.05; ** 0.001 < *p* < 0.01, *** = *p* < 0.001. ^a^ = Correlation between the original data of the characteristics recorded in the F_2_ population. ^b^ = Correlation between characteristics recorded in the F_2_ population and frequencies of desirable character of each characteristic recorded in the F _2:3_ population. ^c^ = Correlation between frequencies of desirable character of each characteristic recorded in the F_3_ population. Note: characters regarded as most desirable for FH, SY, SHR, ER, and PE are determinate flowering, high seed yield per plant, high pod shatter resistance, early maturing, and perennial growth habit.

**Table 4 genes-11-01223-t004:** List of the nine QTLs for number of stems per plant (NS), stem growth orientation (SO), plant height (PH), or perenniality (PE), their respective position on the eight field cress linkage groups (LGs), as well as the genomic region that they span, the flanking SNP markers of each QTL, their LOD score, and the percentage of phenotypic variance that they explained (PVE). The homologous regions in *A. thaliana* chromosomes (AtChr), corresponding to the genomic regions spanned by seven of the nine QTLs, are shown in the last three columns.

Trait	QTL ^a^	LG	Position (cM)	QTL Region (cM)	Flanking Marker Loci ^b^		LOD	PVE (%)	AtChr ^c^	Start (nt)	End (nt)
					Left	Right					
NS	*qNS1-1*	1	17.00	16.13–17.93	sc26535_625A/G	sc21796_243C/T	13.04	11.81	1	6630234	6816262
SO	*qSO2-1*	2	26.00	25.80–27.17	sc33009_557C/A	c1443202_1111C/G	3.21	1.93	NA	NA	NA
PH	*qPH3-1*	3	72.00	71.33–72.85	sc11378_1260A/G	sc23699_989T/G	3.04	3.16	3	4211673	4582690
SO	*qSO4-1*	4	1.00	0.00–1.38	sc9980_604A/T	sc3610_484A/G	3.16	1.84	2	19096856	19679860
PE	*qPE4-1*	4	76.00	75.12–77.94	sc2646_16112C/A	sc16365_3162C/T	3.64	7.23	NA	NA	NA
PH	*qPH6-1*	6	45.00	44.86–46.38	sc28125_3971C/A	sc26361_413G/C	14.89	16.94	5	20689846	20845218
NS	*qNS6-1*	6	77.00	76.48–78.14	sc19636_661A/T	c1445932_1241C/T	5.95	5.06	5	23044365	23311700
PH	*qPH7-1*	7	48.00	47.89–48.08	sc21252_1620C/T	sc20822_9341C/T	8.66	9.39	5	15556187	15708117
NS	*qNS8-1*	8	36.00	31.95–42.77	sc14984_884T/C	sc2744_150835G/A	4.63	4.42	5	9100466	24634605

^a^ In a QTL name, “q” stands for QTL, the two letters following “q” refer to a trait, the number before the hyphen is the linkage group to which the QTL was mapped, and the “1” after a hyphen shows that the QTL was the first for that trait on that linkage group. ^b^ In a marker locus name, the portion before “_” refers to a scaffold (sc) or a contig (c), in which the locus was found, whereas the portion after “_” refers to the position of the SNP in that scaffold or contig, followed by the SNP alleles in that locus, given in the form of “N/N”. ^c^ The GenBank accession numbers of *A. thaliana* chromosomes (AtChr) are CP002684.1 for AtChr 1, CP002685.1 for AtChr 2, CP0002686.1 for AtChr 3, and CP002688.1 for AtChr 5. NA = not applicable.

**Table 5 genes-11-01223-t005:** Percentage of plants within a phenotypic class that are AA or BB homozygote for the left flanking markers of each of the nine QTLs across the four characteristics.

Phenotypic Class	Geno-Type	Characteristic	Number of Stems Per Plant (NS)			*p* Value
		Marker	sc26535_625A/G	sc19636_661A/T	sc14984_884T/C	
		QTL	*qNS1-1*	*qNS6-1*	*qNS8-1*	
≥10 stems	AA		13	4	17	<0.01
	BB		46	29	25	
Single stem	AA		56	28	28	<0.05
	BB		8	13	18	
		Characteristic	Plant Height (PH)			
		QTL	*qPH3-1*	*qPH6-1*	*qPH7-1*	
		Marker	sc11378_1260A/G	sc28125_3971C/A	sc21252_1620C/T	
>30cm	AA		27	3	9	<0.05
	BB		15	48	34	
<20 cm	AA		20	38	44	<0.01
	BB		39	11	11	
		Characteristic	Stem growth orientation (SO)			
		QTL	*qSO2-1*	*qSO4-1*		
		Marker	sc33009_557C/A	sc9980_604A/T		
Erect	AA		15	37		<0.01
	BB		38	18		
Creeping	AA		20	33		<0.01
	BB		27	17		
		Characteristic	Perenniality (PE)			
		QTL	*qPE4-1*			
		Marker	sc2646_16112C/A			
Perennial	AA		14			<0.01
	BB		34			
Biennial	AA		19			<0.05
	BB		24			

Note: the allele whose homozygous genotype is in a higher percentage within a phenotypic class of a given characteristic is favorable to obtain that phenotypic class of that characteristic. Considering only the left flanking markers of the nine QTLs, the following were evident: (1) for the three marker loci of NS QTLs, AA + AA + AA and BB + BB + BB genotypes had 2.5 and 8 stems per plant on average, respectively; (2) for the three marker loci of PH QTLs, the average plant height of AA + BB + BB and BB + AA + AA were 34 and 11 cm, respectively, and 78% of plants with the AA + BB + BB genotype were taller than 30 cm; (3) at the two marker loci of SO QTLs, 77% of plants with BB + AA genotypes were erect. Note: the order of the marker loci combined with “+” are as given, from left to right in the table.

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
