# Peer review of "QTL Mapping for Domestication-Related Characteristics in Field Cress (Lepidium campestre)—A Novel Oil Crop for the Subarctic Region"

_genes, 2020, doi:10.3390/genes11101223_

Round 1
Reviewer 1 Report
I carefully read the manuscript entitled "QTL mapping for domestication related characteristics in field cress (Lepidium campestre) – a novel oil crop for the subarctic region". I believe that this manuscript can be published in Genes after MINOR revision.
Line 6: “Affiliation” should be removed.
Line 32: needs reference(s).
Line 34: needs reference(s).
Line 39: needs reference(s).
Line 43: needs reference(s).
Line 50: needs reference(s).
Line 64: needs reference(s).
Line 100: Greenhouse conditions such as temperature, humidity, light intensity and quality should be clarified.
Table 4 should be presented as a text, not Figure.
More discussion is required especially for QTL mapping.
The rest of the manuscript is presented in a good manner.
Author Response
"Please see the attachment"

Reviewer 2 Report
Summary
In this study, Hammenhag et al., performed quantitative trait locus (QTL) mapping for a suite of agriculturally-relevant traits in an emerging oilseed crop, Lepidium campestre (field cress). They identified a number of QTL hits that explain a substantial proportion of plant architecture and growth habit. To identify putative candidate genes, they compared syntenic regions between the L. campestre QTLs and the model plant, Arabidopsis thaliana, which is a close relative.
Comments
- I suggest simplifying your trait abbreviations. For example, instead of NSPP for number of stems per plant, why not use ST for stems? Similarly, SYPP (seed yield per plant) could be changed to Y (yield). Rather than lending clarity, the complex abbreviations impede readability introducing new or uncommon initialisms that must be remembered.
- Similarly, I suggest that you use the same abbreviations for the QTL prefixes and the traits. The ‘Q’ prefix is not necessary to designate QTLs. Perhaps the two-letter abbreviations you use for the trait associated with each QTL could also serve as the trait abbreviation through the manuscript. This simplification would greatly increase readability.
- I find the term ‘F2:3’ to describe progeny of the F2 population to be confusing. Is an F2:3 individual the same as an F3 population, derived from selfing F2 individuals or is it something more complex? If F2:3 is the same as F3, I suggest you use the latter abbreviation. If it is more complex, describe the crossing scheme in the methods and perhaps include a diagram to illustrate.
- Is Lepidium campestre primarily a selfer? What is the rate of out-crossing? What steps did you take to prevent cross-pollination when generating the F2 and F3 mapping populations?
- Lines 81-83: Cross-pollinating two genotypes yields an F1, not an F2. Please provide more details about how the F2 population was generated. When generating the F1 individuals, what steps were taken to prevent self-pollination? Which genotype served as the pollen donor and which the pollen acceptor? To create the F2 population, did you collect seeds from one or several self-pollinated F1 plants?
- Lines 98-99: Please be precise here: replace ‘some other’ with the exact number of genotypes planted.
- Lines 101-102: Your numbers do not add up here: 375 + 2 + 4 = 381, not 380. Please correct this error.
- Lines 127-132: Based on your description here, the plants that you designate as biennial could more appropriately be called annual. Plants were germinated in August 2016, flowered in spring of 2017, and were harvested in summer and autumn of 2017, which represents just one year. Plants that fully died out after that first harvest (i.e. completed their life cycle in one year) would be considered annuals. On the other hand, biennials generally take two years to complete their life cycle. Please clarify why you use the term biennial and/or adjust your nomenclature.
- Lines 141-143: Please describe how the sequencing data was processed and analyzed. Were reads aligned to a reference genome? How were SNPs identified? What tools were used?
- Figure 1: It is not clear to me what value this figure holds. Rather than pie charts documenting the overlap in desired traits in the F3 individuals, sketches or pictures of select plants illustrating the desirable and undesirable levels of different traits would be useful.
- Figure 2: Please make the font in this figure larger, especially on the LOD score plot.
